# An Environmental Niche Model to Estimate the Potential Presence of Venezuelan Equine Encephalitis Virus in Costa Rica

**DOI:** 10.3390/ijerph18010227

**Published:** 2020-12-30

**Authors:** Bernal León, Carlos Jiménez-Sánchez, Mónica Retamosa-Izaguirre

**Affiliations:** 1Biosecurity Laboratory, Veterinary Service National Laboratory, Animal Health National Service, Ministry of Agriculture and Cattle, Heredia 40104, Costa Rica; 2Laboratory of Virology, Tropical Diseases Research Program (PIET), School of Veterinary Medicine, National University, Heredia 40101, Costa Rica; carlos.jimenez.sanchez@una.ac.cr; 3International Institute for Wildlife Conservation and Management, National University of Costa Rica, Heredia 40101, Costa Rica; mretamos@una.cr

**Keywords:** VEEV, predicted map, MaxEnt, Costa Rica, arbovirus, zoonotic

## Abstract

Venezuelan equine encephalitis virus (VEEV) is an arbovirus transmitted by arthropods, widely distributed in the Americas that, depending on the subtype, can produce outbreaks or yearly cases of encephalitis in horses and humans. The symptoms are similar to those caused by dengue virus and in the worst-case scenario, involve encephalitis, and death. MaxEnt is software that uses climatological, geographical, and occurrence data of a particular species to create a model to estimate possible niches that could have these favorable conditions. We used MaxEnt with a total of 188 registers of VEEV presence, and 20 variables, (19 bioclimatological plus altitude) to determine the niches promising for the presence of VEEV. The area under the ROC curve (AUC) value for the model with all variables was 0.80 for the training data and 0.72 for the test. The variables with the highest contribution to the model were Bio11 (mean temperature of the coldest quarter) 32.5%, Bio17 (precipitation of the driest quarter) 16.9%, Bio2 (annual mean temperature) 15.1%, altitude (m.a.s.l) 6.6%, and Bio18 (precipitation of the warmest quarter) 6.2%. The product of this research will be useful under the one health scheme to animal and human health authorities to forecast areas with high propensity for VEEV cases in the future.

## 1. Introduction

The genus Alphavirus is constituted by 31 species of virus. One of the most important members of this genus in the Americas is the Venezuelan equine encephalitis virus (VEEV) [1]. VEEV is transmitted by arthropods, mainly mosquitoes [2,3], and can infect not only equines but also different species, including humans [4,5]. There are three fundamental factors that keep arboviruses circulating and producing outbreaks in a given geographic area: virus, vectors, and the host-reservoirs. These components must be present in the same place and at the same time for the effective transmission of pathogens to occur and many variables influence or affect each of these components. VEEV is closely related to its mosquito vectors, and, these, in turn, are intently related to weather conditions. Vectors are influenced by climatic variables such as temperature, humidity, and rain [6], as well as elevation [7] or by less ecological aspects such as transportation of larvae or adults to new areas where they did not exist before through vehicles or objects [8]. The reservoirs and hosts maintain the viruses both in the enzootic and occasionally in the epizootic cycles. Host factors such as susceptibility and permissibility favor the replication of these viruses [9].

The first report of an outbreak of VEEV in Costa Rica was documented in 1970, and this study also gave evidence indicating that subtype IE is endemic in the country [2]. An IgG seroprevalence study carried out in 2013 demonstrated that VEEV was circulating in the lowlands, less than 900 meters above sea level (m.a.s.l) and highlands (more than 900 m.a.s.l), with an overall seroprevalence of 35.9% (in 217 horses) [10]. In that study, altitude <100 m was the only variable considered a risk factor in the multivariate analysis, indicated that the lower altitude the higher IgG positives cases to VEEV [10]. An 11 years passive surveillance study finished in 2019, (in which animal health authorities did not actively search for cases but the samples were sent to the laboratory voluntarily by the animal owners), showed the presence of positive cases every year with apparent peaks every six years, one observed in 2009 and the second in 2015 during the evaluated period. Altitude was associated with positive cases, consistent with the previous cross-sectional study [11]. However, until now there is no information about potential places where this virus can infect horses or humans. An ecological niche is the sum of all the environmental factors acting on the particular organism that occupies a specific area or subdivision of the habitat [12]. Ecological niche models calculated with presence/absence, presence and pseudo-absence, or presence-only data can be therefore considered representations of different realized niches [12]. However, the popularity of presence-only niche models has arisen in part due to the increased availability of presence-only records, especially if we only have information about clinical cases produced by a particular virus, as in our situation. MaxEnt utilizes a maximum entropy algorithm to analyze the values of environmental layers. In other words, the program analyzes the variables and choose the ones of maximum entropy, i.e., the most unconstrained ones. This machine learning program uses only data for the presence of a particular specimen, in addition to climatological and geographical variables, across a user-defined landscape that is divided into grid cells to obtain a solution, comparing the conditions where these cases are present to model possible niches in other regions that could have these favorable conditions and estimate the suitability of presence of new cases.

Based on these reasons Maxent was selected to model the spatial data based on the serological presence of VEEV in horses to establish potential areas at risk for the appearance of this virus in Costa Rica using bioclimatic variables, and elevation data. The knowledge generated with this study could be important to help to prioritize resources and improve planning, prevention, and response strategies to future surveillance and control programs for this virus.

## 2. Materials and Methods 

### 2.1. Study Area

Costa Rica is a small tropical country located in Central America with a land area of 51,100 km^2^. It lies between 8° and 11°15′ north latitude, and between 82° and 86° west longitude, with Nicaragua to the north and Panama to the south-east. The Pacific Ocean is on its west coast, and the Caribbean Sea on its east coast. Altitude ranges from sea level on the Pacific and Caribbean coasts up to 3819 m.a.s.l in the mountain region [13]. Costa Rica is administratively divided into seven provinces and 82 cantons [13].

### 2.2. Input Data

We included two sets of VEEV positive data confirmed by two methodologies; the plaque reduction neutralization test (PRNT) and the IgM Enzyme-Linked ImmunoSorbent Assay ELISA from two different serological studies. In the first study, 217 horses were randomly selected across the country in the lowlands and highlands, 81 horses were IgG positive by ELISA and confirmed by PRNT [10]. The second study comprised passive surveillance carried out between 2009 and 2019 involving samples from animals with neurological signs that were taken and sent to the laboratory at the moment the animals present symptomatology to detect the presence of IgM antibodies to VEEV and other arboviruses. One-hundred-and twenty-eight horses had IgM antibodies to VEEV but only 107 had geographic coordinate data [11]. One of these cases was also positive by RT-PCR, [14]. In both studies, VEEV was the virus more prevalent, and only data with coordinates were used for this analysis. Considering these conditions 188 registers of VEEV presence were utilized in this study. Geographic coordinates in both studies were transformed from WGS84 to CRTM05 with an online Coordinate Converter https://tool-online.com/es/conversion-coordenadas.php. The predictor variables were downloaded from Worldclim site version 1.4 23 February 2016 https://worldclim.org/data/v1.4/worldclim14.html. The 19 bioclimatic descriptions are presented in the Table 1. Altitude variable was added to the analysis, which was obtained from the ATLAS of Costa Rica [15], these variables were converted to ESRI ASCII format with Qgis Madeira 3.4.15, long term version. The resolution of the cell was 200 m and 32-bit float pixel type.

#### Environmental Niche Model

MaxEnt 3.3.3k modeling program [16] was utilized to model the distribution of VEEV in Costa Rica based on previously obtained geographical locations. The receiver operating curve (ROC) method was used to assess the overall model predictive performance, a measure of the ability of the model to distinguish presence from the background with a value of 1 indicating a perfect prediction while 0.5 is as good as a random prediction [17] MaxEnt maximizes the so-called gain function, a penalized maximum likelihood function, to find a model that can best differentiate presences from background locations. The gain is closely related to deviance, a measure of goodness of fit used in generalized additive and generalized linear models. It starts at 0 and increases towards an asymptote during the run. During this process, MaxEnt is generating a probability distribution over pixels in the grid, starting from the uniform distribution and repeatedly improving the fit to the data. Maxent isn’t directly calculating the “probability of occurrence”. The probability it assigns to each pixel is typically very small, as the values must sum to 1 over all the pixels in the grid. Then the program uses the presence of the variant of interest to produce a characteristic map that shows suitability values between 0 and 1 indicating high suitability versus low suitability of species presence, respectively, represented by a cloglog format. This range of values is depicted using a color gradient [18,19].

To assess the output map format, the results of the logistic output were compared with the cloglog [20]. The jackknife test was used to evaluate the environmental predictors individually. Variables are considered important if they produce high training gains when used alone in a model. A variable is also important if the training gain is low when the variable is removed from the model [16]. Of the 188 registers of presence 25%, (47 registers), were used for testing purposes (a threshold to make a binary prediction). To determine the best combination of environmental data for modeling, the model was run twice using different sets of input layers each time: (1) all layers (bioclimatic and altitude variables) (2) only bioclimatic layers (altitude was removed) [21].

Finally, to appraise the output plot produced by MaxEnt, a group of 8 IgM VEEV positive horse cases reported in April-2016 [22], was located in this plot using Qgis software to observe if these cases fit with the niches predicted by MaxEnt.

## 3. Results

A total of 141 data were analysed in the training model The AUC value for the model with all variables was 0.80 for the training data and 0.72 for the test. When only the climate variables were used the AUC was 0.78 for the training data and 0.72 for the test data.

Table 1, presents estimates of relative contributions of the environmental variables to the MaxEnt model and shows a comparison of the effect of altitude over the climatology variables.

The variables with the highest contribution (in percentages) to the model when all the variables were included were Bio11 (mean temperature of the coldest quarter) 32.5%, Bio17 (precipitation of the driest quarter) 16.9%, Bio2 (annual mean temperature) 15.1%, altitude (m.a.s.l) 6.6% and Bio18 (precipitation of the warmest quarter) 6.2%.

The variables mean temperature of the coldest quarter and precipitation of the driest quarter contribute 49% of the VEEV presence cases, while the remaining three variables annual mean temperature, altitude and precipitation of the warmest quarter, provide a 28.2% contribution to the model. How each one of these variables affects the suitability of VEEV presence is detailed in the response curves shown in Figure 1 and Figure 2.

Bio 11 (Figure 1A) shows how the suitability of the presence of VEEV increases when the temperature is higher than 25 °C in niches with temperatures in the coldest quarter of the year (last months of the year, reaching 80% of the suitability of presence). However, when the Mean temperature of the coldest quarter is analysed alone without the effects of the other variables (Figure 2A), the suitability increases after 25 °C and is sustained with a little decrease.

Precipitation of the driest quarter is an index that provides the total precipitation during the driest three months of the year (Figure 1B). The suitability of VEEV presence increases with the first rainfalls and keeps going up until reaching 450 mm of precipitation and from there the curve begins to flatten with a 100% suitability of VEEV presence. Interestingly, when only this variable is analysed without the effects of other variables, the unique necessary condition to increase the suitability of VEEV cases (from 34% to 94%), is the presence of the first rainfalls. However, this likelihood falls to 45% very soon, indicating that perhaps other conditions are necessary to keep a high number of VEEV cases (Figure 2B).

The annual mean temperature (Figure 1C) is the annual temperature change. The opportunity of the presence of VEEV increases just over 50%, until a 10 °C temperature change occurred in the year. When this variable is analysed alone, the suitability of VEEV presence was almost 80% (Figure 2C).

While the suitability of VEEV presence increases to 80%, with increasing altitude, and it drops as altitude reaches 1400 m.a.s.l, (Figure 1D).

However, when this variable is analysed alone, the odds decreased when the altitude is higher than 100 m.a.s.l (Figure 2D).

Precipitation of the warmest quarter is diluted in the presence of other variables over 50% after 200 mm of rain (Figure 1E). Its effect alone increases the chance of VEEV presence to 85% after 200 mm of rain during the warmest quarter (Figure 2E).

An alternative test that estimates which variables are most important in the model is presented in the jackknife test (Figure 3). Mean temperature of the coldest quarter is the variable with the highest gain ~0.21 when it is used in isolation and therefore appears to have the most useful information by itself, (Figure 3A), in agreement with the results shown in Table 1. While precipitation of the warmest quarter is also the most effective single variable for predicting the distribution of the occurrence data that was set for testing when predictive performance is measured using AUC 0.72 (Figure 3B).

To demonstrate the suitability of this map, positive VEEV equine cases of the 2016 outbreak located in Talamanca Sixaola, (the southeastern region of the country) were also represented in Figure 4 These cases were not part of the database used in the MaxEnt analysis and were added to the MaxEnt plot map using the Qgis program. The site where the cases were diagnosed has values that range from 62 to 77% of VEEV suitability according to the MaxEnt model. Finally, Figure 5 presents the incidence of VEEV outbreaks or cases for each of the 82 counties of Costa Rica, where the label color in the map indicates the suitability of VEEV cases.

## 4. Discussion

VEEV is a neglected disease in Latin American countries. In Costa Rica few cases of people infected with this virus have been reported, despite the fact that based on serosurveys 36% of the horses have been in contact with this virus and cases are widely distributed throughout the country [10]. For this reason, it is important to determine the possible niches where cases could appear in the future.

MaxEnt is a software package widely applicable in the ecology field. It is used to model the suitability of the presence of a particular species based on the distribution of occurrence records in a given niche, combining environmental variables such as rainfall, temperature, and geographical layers to evaluate the study’s areas.

These conditions make it the perfect solution to establish possible sites where these species can be found, be it an endangered species or an infectious agent. The positive data (presence) used in this analysis came from two different serological studies, one done in 2013 based on the presence of IgG Alphavirus antibodies by ELISA. Due to the IgG cross-reaction between alphavirus species, the results were confirmed by PRNT to avoid false positives (commission errors), as requirements to include these animals in that study, the horses must be not vaccinated against alphavirus or have been moved from premises before the samples were taken (omission errors). In this study we are detecting IgG antibodies, positive animals to VEEV, but we don’t know when exactly they were infected. For example, we have positive and negative cases from the same area, with the same weather or variable conditions, but probably some of the negative animals are younger than the positive (older animals have more time to be exposed and infected by the virus). If we use negative samples as absence data in another kind of model, we can commit omission errors. If a virus has a low prevalence, you need a big size sample to be sure that the disease is absent in an area and this sampling is expensive, and we are prone to make omission errors [23], The rest of the location data came from a second serological study, based on the determination of IgM antibody by ELISA. When an animal presented any nervous symptoms a sera sample was taken and sent to the laboratory to determine the presence of IgM antibody against different arboviruses. IgM antibodies are detectable in acute infections for up to 6 weeks [24]. Though, IgM antibodies from an alphavirus response could last around 2 to 3 months [25]. At the difference to IgG, IgM antibody seems to be more specific than IgG [11,26], then the positive sample was not confirmed by another test. However, in this study, the symptomatology could have been caused by other infectious agents or even a toxicological etiology, as happened in 2019 with 141 cases but only six were positive for VEEV, then the negative data did not correspond to real absence data and could affect the predicting results. For these reasons the EMN based on presence, like MaxEnt, is a perfect tool to analyze our VEEV data, where we are sure we can trust the positive cases but not necessarily in the negative cases.

The ability of a model to predict depends on how well the data fits the training model, as well as the kind of variables used in the model, and suitability value is represented by training AUC.

Even though the training AUC value in this study could be considered low (0.80) in comparison with other studies, (Ebola virus presence AUC 0.9 [27] or mosquitoes vector presence 0.94 and 0.9 [21,28]), however, it was similar to that reported in another model based on the presence of Eastern Equine Encephalitis virus (EEEV), another Alphavirus species, 0.770 in the training data and 0.758 for the test data [29] AUC of 0.8 are considered from acceptable to excellent [16]. In this study, the test data was 0.72, both AUC values exceed a threshold value of 0.5, which corresponds to the AUC model generated at random.

The response curves demonstrate that the presence of VEEV is linked to several variables; five of them have the highest value of the contribution to the model, two were related to temperature (1st and 3rd positions), two with precipitation (2nd and 4th), and altitude. It is remarkable that mean temperature of the coldest quarter was the variable that contributes the most to explain the presence of VEEV, considering our results mean temperature greater than 25 °C. In Costa Rica the months of November, December and January are the coldest of the year, and according to a VEEV surveillance study done in Costa Rica from 2009 to 2017, the months of October had 19% of the total of the positive cases, while November had 29% of the positive cases, these months had the highest cases of VEEV during the studied period [22].

Temperatures between 25 to 27 °C, prolong the life span of *Aedes taeniorhynchus* [30] and also enhance the spread of VEEV from the midgut to the hemocoel more rapidly [31] allowing virus transmission. While at 24.4 °C the larval cycle of *Deinocerites pseudes* lasted 3 or 4 weeks from hatching to pupation and 3 or 4 days in the pupal stage [32]. *Ae. taeniorhynchus* and *Deinocerites pseudes* are two mosquito species that transmit VEEV in Costa Rica [2,32].

During the driest quarter (January to March), the chance of VEEV presence increases from 30% with the first rainfall to 100% when the rainfall increased to 600 mm. But when Precipitation of the driest quarter was analysed alone, the feasibility of VEEV presence goes up 90% just after the first precipitations, indicating that rainfall is probably the variable that triggers the conditions necessary to produce VEEV outbreaks. It has been described that in months with more than 200 mm of precipitation, mosquitoes populations increase, [5], raising the suitability of outbreaks or cases, however other variables are required to maintain a high suitability of VEEV cases (Figure 1 and Figure 2B). The mean temperature of the coldest quarter is also the variable with the highest training gain according to the Jackknife test (Figure 3). Temperature and precipitation variables have an important role in the presence of VEEV, although altitude is not a climatological variable it is correlated to precipitation and temperature.

Figure 2D shows how VEEV presence increases up to 80% with the altitude and then falls after reaching 1400 m.a.s.l, however, the incidence decreased when the altitude is higher than 100 m.a.s.l when the variable is analysed alone without the contribution of other variables. This means that other conditions favor the occurrence of VEEV at higher elevations; one driver is the presence of different vector/reservoir species that may be more or less abundant. It was expected that the altitude variable would contribute more to the VEEV presence model since it was the one risk factor associated with antibodies to VEEV in the Costa Rican study [10]. Altitude also appeared to be the most important variable reported in an EEEV study [29], or for results reported for the vectors *Culex. tritaeniorhynchus* [21], and *Aedes albopictus* [28].

It is clear that many variables interact and that other variables not included in this study could have an important role in the presence of VEEV cases. These could include land cover, distribution of horses, and probably the most significant, the presence of VEEV vectors and reservoirs. The latter is dependent on the existence of an appropriate number of infected reservoirs required to infect vectors that then transmit the virus to an appropriate host. *Sigmodon hispidus* (cotton rat) is an enzootic reservoir of VEEV [33], which is widely distributed from the USA to Venezuela [34]. This rat is more abundant at lower elevations with higher temperatures and in areas of more sparse vegetative cover [35]. The reported habitat of the vector *Aedes. taeniorhynchus* is up to 800 m.a.s.l [36], however, there are other reservoirs and vectors capable of sustaining and replicating the virus and transmitting it to hosts such as horses or animals, who must be able to live at a higher altitude.

MaxEnt has demonstrated to be extremely useful for species conservation but also allows epidemiologists and health workers to determine places with ideal conditions for the presence of new cases of etiological agents. The map in Figure 5 could be used in future studies to place mosquito traps to monitor the species of mosquitoes that test positive for the presence of VEEV or even to set sentinel animals.

## 5. Conclusions

Based on the available variables, temperature, precipitation and elevation have an important impact in the presence of VEEV cases in Costa Rica, probably associated with its effect on the biological cycle of its vectors and reservoirs. We hope this information could be useful, under the one health scheme, to animal and human health authorities to know where VEEV cases could appear in the upcoming years, and make decisions about how to manage this virus in the future as well as to monitor the counties with more chance of having either horse or human cases.

## Figures and Tables

**Figure 1 ijerph-18-00227-f001:**
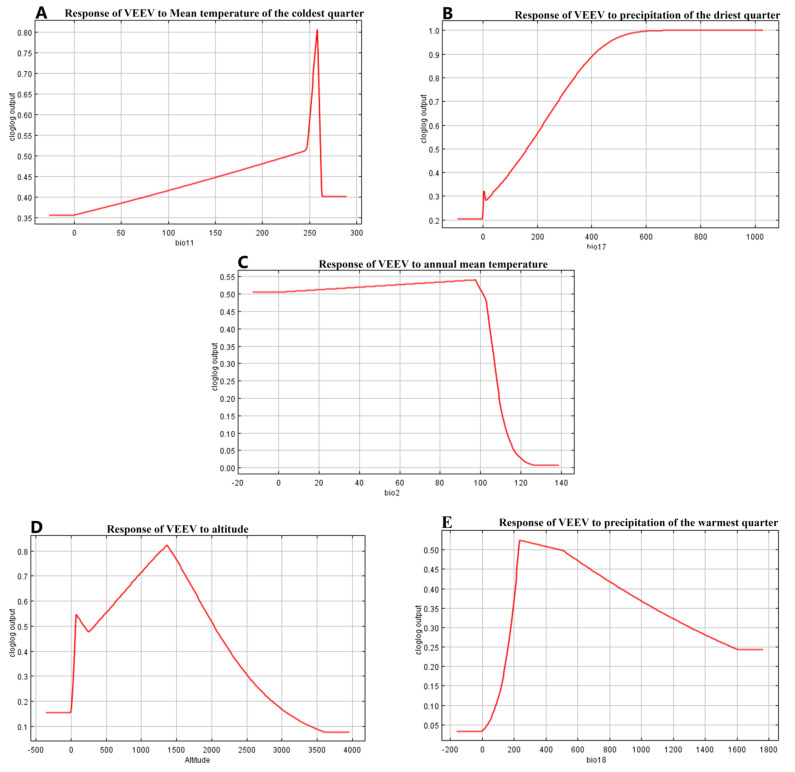
Shows the effect of each individual variable, that most contribute to the suitability of VEEV cases, when the remaining variables are configured to their average value. In (**A**), Bio11 Mean Temperature of the coldest quarter, the *X*-axis represents Degrees Celsius by 10, (**B**) Bio17 Precipitation of the driest quarter the units are millimeters of precipitation, (**C**), Bio 2 Annual mean temperature the units are Degrees Celsius by 10, (**D**) *X*-axis units are millimeters above sea level, and (**E**) units are millimeters of precipitation.

**Figure 2 ijerph-18-00227-f002:**
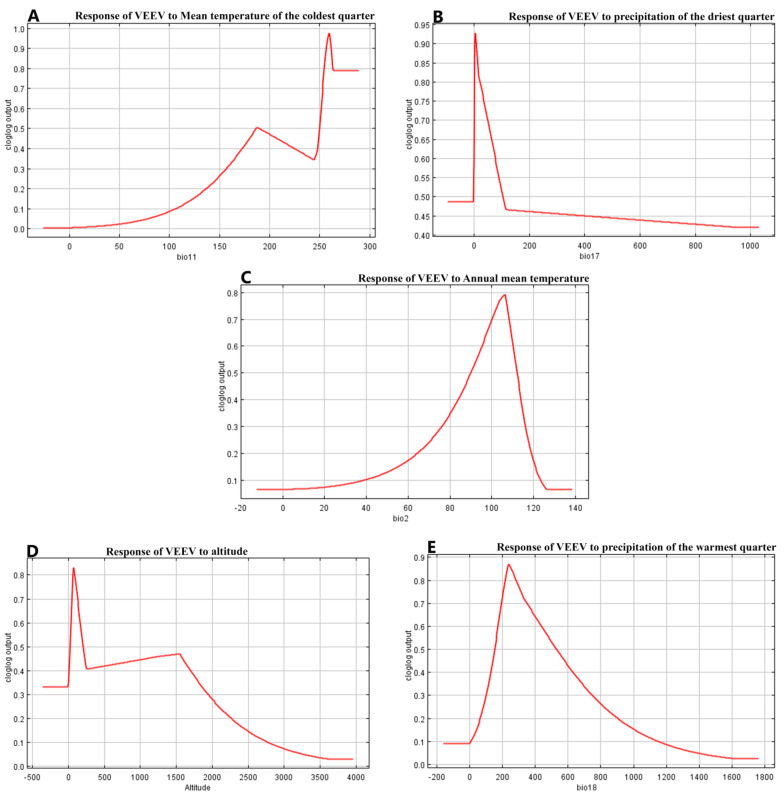
Depicts the effect of each of the variables that contribute the most to the model analyzing each variable by itself without the interaction of other variables. (**A**), Bio11 Mean Temperature of the coldest quarter, the *X*-axis represents Degrees Celsius by 10, (**B**) Bio17 Precipitation of the driest quarter the units are millimeters of precipitation, (**C**), Bio 2 Annual mean temperature the units are Degrees Celsius by 10, (**D**) *X*-axis units are millimeters above sea level, and (**E**) units are millimeters of precipitation.

**Figure 3 ijerph-18-00227-f003:**
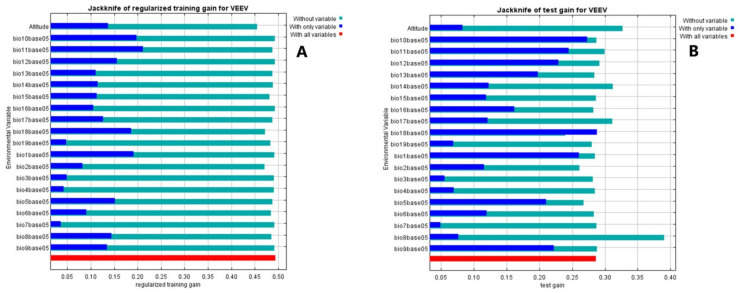
In the Jackknife test, a model is created using each variable in isolation and also all the variables. The blue bars show the effect of each variable over the model by itself, the light blue bars the effect in the model when this variable is not considered. The red bar represents the performance of the model when all variables are included, if a blue light bar (the model is not using this variable) is longer than the red bar, means that the predictive performance of the model improves when the corresponding variable is not used. In our case, none variable. (**A**), shows the training gain for VEEV (the full model), while (**B**) shows the test gain for VEEV (the effect of variables in the set of samples used to validate the model). The maps of the logistic and cloglog outputs are shown in the Appendix A. Whereas the map of the MaxEnt model for VEEV using the cloglog output, (Figure 4), depicts areas with higher suitability conditions for VEEV presence in warmer colors (orange-red). Black dots show the presence locations used for training, while violet dots show the test locations. The blue color represents the region with a low odd of VEEV presence.

**Figure 4 ijerph-18-00227-f004:**
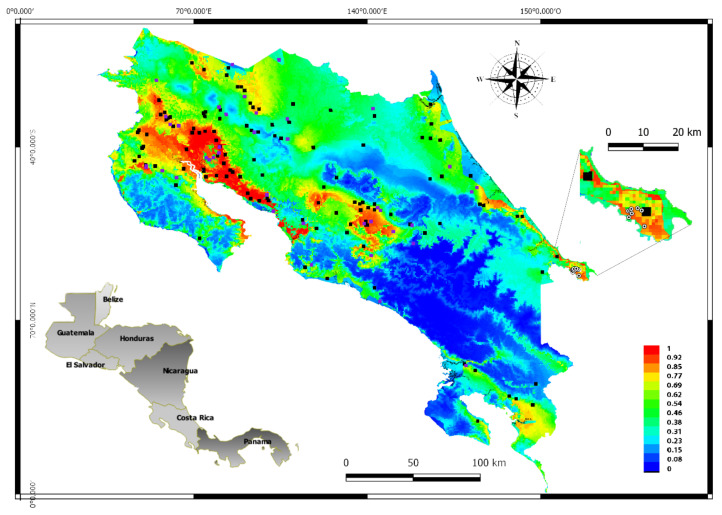
The main map represents the MaxEnt output, the black dots are the VEEV positive data used in the training analysis, while the violet dots are the VEEV positive data used in the test analysis. The colors in the map indicate the suitability of the VEEV presence, the warmest color (yellow to red) shows the places with more suitability of VEEV presence, (0.7 to 1) while the coldest colors (deep blue to light green shows the lower suitability of VEEV presence 0 to 0.69, see the scale color. The amplified region (Sixaola) south of the country shows the cases of VEEV in 2016, which were not part of the analysed data in this study, the white and black dots shows the locations of VEEV positive cases by IgM ELISA [11]. The map in gray represents the location of Costa Rica in Central America.

**Figure 5 ijerph-18-00227-f005:**
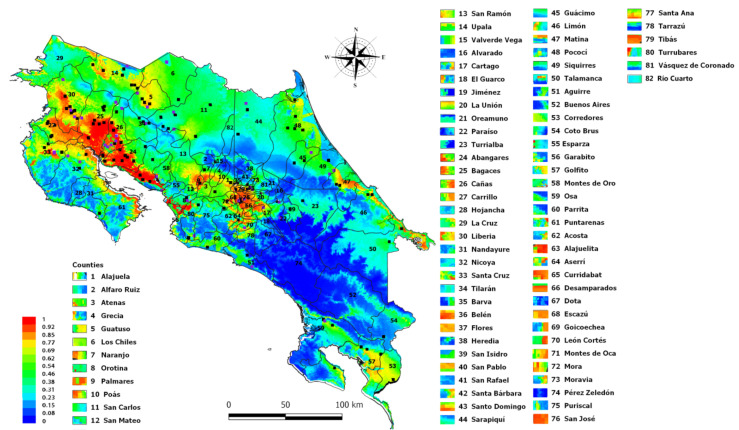
MaxEnt output map represented the suitability of VEEV outbreaks by each of the 82 Costa Rica’s counties., The rectangles next to the county name (labels) represent the potential risk of VEEV presence in that county.

**Table 1 ijerph-18-00227-t001:** Worldclim 19 variables description plus altitude.

Variable	Description of Worldclim 19 Variable Data Represented from 1960–1990, Altitude and Coverage	Contribution % All Variables	Contribution % Only Climatological Variables
Bio01	Annual mean temperature, °C	0.3	3.1
Bio02	Mean diurnal range (Mean of monthly (max temp-min temp)), °C	**15.1**	**19.2**
Bio03	Isothermality [(Bio2/Bio7) * 100], °C	0.8	1
Bio04	Temperature Seasonality (standard deviation * 100), °C	0	0.4
Bio05	Max temperature of the warmest month, °C	1.7	1.7
Bio06	Min temperature of the coldest month, °C	4.7	4.9
Bio07	Temperature annual range (Bio5-Bio6), °C	3.3	2.5
Bio08	Mean temperature of the wettest quarter1, °C	3.4	2
Bio09	Mean temperature of the driest quarter, °C	0.1	0.2
Bio10	Mean temperature of the warmest quarter, °C	0.4	1.3
Bio11	Mean temperature of the coldest quarter, °C	**32.5**	**27.3**
Bio12	Annual precipitation, mm	0	0.2
Bio13	Precipitation of the wettest month, mm	0.6	0.6
Bio14	Precipitation of the driest month, mm	0.6	1.9
Bio15	Precipitation seasonality (coefficient of variation), mm	3.1	3.1
Bio16	Precipitation of the wettest quarter, mm	0.6	0.1
Bio17	Precipitation of the driest quarter, mm	**16.9**	**18.8**
Bio18	Precipitation of the warmest quarter, mm	**6.2**	**8.6**
Bio19	Precipitation of the coldest quarter, mm	3.8	4.1
Altitude	Meters about sea level	**6.6**	-

In bold are indicate the variables with higher contribution to the model.

## Data Availability

Data available upon request.

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
