# Peer review of "An Environmental Niche Model to Estimate the Potential Presence of Venezuelan Equine Encephalitis Virus in Costa Rica"

_ijerph, 2020, doi:10.3390/ijerph18010227_

Round 1

Reviewer 1 Report

Leon et al. describe a predictive model to identify potential areas at risk for VEEV outbreaks, which could have a significant impact on surveillance and outbreak prevention. This model could provide a basis for allocation of resources that could prevent VEEV zoonosis. However, the AUC values presented here (0.8) are not high enough to be useful as a predictive model.

Introduction

The introduction should be enhanced. Please provide an elaboration on current modeling methodologies that have been applied to VEEV predictions and the associated limitations. Then explain how MaxEnt compares to these other methods.

Materials and Methods

Divide the “input data” section into subsections in order to clearly describe the various components of the manuscript (i.e. input data, worldclim analysis, model description and jackknife test).

L73: Correct “complete sequenced”

Add section to describe statistical analysis. How is error and statistical significance calculated?

Results

Table 1: Are the percentage values means? Error values should also be included. How were the bold values selected? Was there a statistical analysis?

L105-112: Can error values be added to the percentages described here?

Figures 1 and 2: It would be easier to visualize and follow the results text descriptions if these figures were combined for example to show Fig1A side-by-side with Fig2A to easily compare bio11 responses with and without other variables. Also, please label the text throughout and in the figure with the actual parameter instead of the bio number. It is also unclear what the x-axis depicts so please clarify. Also, what are the variables that are interacting in Figure 1? Are all of the variables described in Table 1 being used?

L115: Fix the word “configured”

L132: delete “to” before the word reaching

Be consistent with the spacing and punctuation. Example: L138 “Figure 1 c” as compared to L140 “Figure 2c”.

Figure 3: The panels should be labeled “A” and “B”. Change L152 from “Figure 3 right” to “Figure 3b”. Clearly describe the difference between panels A and B. Were there any statistical analysis performed?

Figure 4: Please include units to go with the color key in the figure and a description of the colors in the figure legend.

L169 and 173 (as well as L177 and throughout the manuscript) the words suitableness and suitability are used to describe VEEV cases. Please clarify this terminology. Do you mean high likelihood or probability?

L177: Correct “Maxent”

Discussion

The reference to bio numbers instead of the actual parameter makes this section difficult to understand.

Conclusions

L249: The term “preponderant protagonist” is very confusing.

Author Response

Introduction

The introduction should be enhanced. Please provide an elaboration on current modeling methodologies that have been applied to VEEV predictions and the associated limitations. Then explain how MaxEnt compares to these other methods.

 R/ Changes were done in the introduction please see lines 33-43, 45-56, 69-72, to our knowledge, none model of predicting presence of VEEV using MaxEnt or other program have been published. However, is mention in the discussion, a study with Eastern equine encephalitis virus EEEV a virus, which belong to the same genus but a different species, that was analysed with MaxEnt. 

Materials and Methods

Divide the “input data” section into subsections in order to clearly describe the various components of the manuscript (i.e. input data, worldclim analysis, model description and jackknife test).

L73: Correct “complete sequenced”

R / This was deleted, by a suggestion of another reviewer

Add section to describe statistical analysis. How is error and statistical significance calculated?

Results

Table 1: Are the percentage values means? Error values should also be included. How were the bold values selected? Was there a statistical analysis?

 R/ Of the total of contribution  (100 %),  the mathematical algorithm implemented in MaxEnt distributes x percentage of this contribution between the variables, according to the  rating of each variable to the explain the model.  In our case, VEEV potential presence in niches with the same condition environmental conditions found in the positive cases. Please see [2]

L105-112: Can error values be added to the percentages described here?

When MaxEnt finish its analysis, an input is generated, no such errors are present in this input, please see [3]   

Figures 1 and 2: It would be easier to visualize and follow the results text descriptions if these figures were combined for example to show Fig1A side-by-side with Fig2A to easily compare bio11 responses with and without other variables. Also, please label the text throughout and in the figure with the actual parameter instead of the bio number. It is also unclear what the x-axis depicts so please clarify. Also, what are the variables that are interacting in Figure 1? Are all of the variables described in Table 1 being used?

R/ most of the suggestion were done, please see lines 160-168, if we merge figure 1 and 2, the resultant figure will be huge and we prefer not to change them.

 What are the variables that are interacting in Figure 1? Are all of the variables described in Table 1 being used?

R/ The variables of figure 1 and 2 are the variables that contribute the most to the model according to table 1..Please see lines 150-158

L115: Fix the word “configured”

R/It was fixed line 154

L132: delete “to” before the word reaching

R/ deleted line 175

Be consistent with the spacing and punctuation. Example: L138 “Figure 1 c” as compared to L140 “Figure 2c”.

R/Fixed were done in lines 171, 181, and 186.

Figure 3: The panels should be labeled “A” and “B”. Change L152 from “Figure 3 right” to “Figure 3b”. Clearly describe the difference between panels A and B. Were there any statistical analysis performed?

R/ the change was done, please see line 206-216

Figure 4: Please include units to go with the color key in the figure and a description of the colors in the figure legend.

R/ Information suggested was added lines 222-231

L169 and 173 (as well as L177 and throughout the manuscript) the words suitableness and suitability are used to describe VEEV cases. Please clarify this terminology. Do you mean high likelihood or probability?

R/ Maxent did not create a probability value, is not possible to estimate occurrence probabilities from Presence*background data, Please see the following article [4]

L177: Correct “Maxent”

R/ was fixed line 232

Discussion

The reference to bio numbers instead of the actual parameter makes this section difficult to understand.

R/ Changes were done through the manuscript

Conclusions

L249: The term “preponderant protagonist” is very confusing.

R/ In lines 336-337 they were replaced by an important impact

  1. Elith, J.; Phillips, S.J.; Hastie, T.; Dudík, M.; Chee, Y.E.; Yates, C.J. A statistical explanation of MaxEnt for ecologists. Divers. Distrib. 2011, 17, 43–57, doi:10.1111/j.1472-4642.2010.00725.x.
  2. Phillips, S.J.; Anderson, R.P.; Dudík, M.; Schapire, R.E.; Blair, M.E. Opening the black box: an open-source release of Maxent. Ecography (Cop.). 2017, 40, 887–893, doi:10.1111/ecog.03049.
  3. Phillips, S.J. A Brief Tutorial on Maxent. Lessons Conserv. 2017, 49, 633–641.
  4. Guillera-Arroita, G.; Lahoz-Monfort, J.J.; Elith, J. Maxent is not a presence-absence method: A comment on Thibaud et al. Methods Ecol. Evol. 2014, 5, 1192–1197, doi:10.1111/2041-210X.12252.

Reviewer 2 Report

The manuscript “An Ecological niche model to estimate the potential presence of Venezuelan equine encephalitis virus in Costa Rica” by Leon et al. uses a species distribution modelling approach to access the risk areas for VEEV circulation in the country. The study in general is interesting. However, more information is required to understand why the authors used the presence-only-modelling approach. I also miss a critical discussion of the database, e.g. is the location of the VEEV-positive horses reliable (is the site of diagnostic also the site of infection?)?

Comments:

# l. 31: which kind of arthropods?

# l. 37-38: this means increasing cases in low altitudes? Please rephrase and link to sentence before.

# l. 38: Passive surveillance of? Please clarify.

# l. 39: how do you see the recurring pattern of six years if only analysing 11 years?

# l. 39-40: here temperature and altitude are relevant factors, previously only altitude? Please merge this statements.

# l. 43: “could produce” à Please rephrase. e.g “can infect”

# l. 49: I do not understand. You also have absence-data from the randomly selected horses?

# l. 49: Please give a more comprehensive introduction on the methodological background of maxent.

# l. 73: “and the virus genome was complete sequenced” à not important, delete

# l. 89-90: Random test percentage? Please rephrase this sentence.

# l. 89-90: Please also use the absence data from the study [5] to check for the suitability for absence prediction.

# l. 97-100: Are this the results of one run? If so, please repeat the Maxent runs with changing training and test points to calculate an average AUC.

# l. 105: “suitability of obtaining”? à unclear, please rephrase, do you mean suitability of transmission?

# l. 136: to keep highest option? à unclear, please rephrase

# l. 169: “suitableness”?

# l. 169: Please add all information relevant to understand all details the figures in the figure captions.

# l. 228: Cx. tritaeniorhynchus

# l. 239: Opened the door? Please rephrase.

# l. 245-247: What is the current monitoring scheme?

Author Response

The manuscript “An Ecological niche model to estimate the potential presence of Venezuelan equine encephalitis virus in Costa Rica” by Leon et al. uses a species distribution modelling approach to access the risk areas for VEEV circulation in the country. The study in general is interesting. However, more information is required to understand why the authors used the presence-only-modelling approach. I also miss a critical discussion of the database, e.g. is the location of the VEEV-positive horses reliable (is the site of diagnostic also the site of infection?)?

 R/ The following text was added in the material and methods lines: 99-101

In both studies, VEEV was the virus more prevalent, and only data with coordinates were used for this analysis, Considering these conditions 188 registers of VEEV presence were utilized in this study.

And in the discussion in lines: 256-281

These conditions make it the perfect solution to establish possible sites where these species can be found, be it an endangered species or an infectious agent.  The positive data (presence) used in this analysis came from two different serological studies, one done in 2013 based on the presence of IgG Aphavirus antibodies by ELISA however due to the IgG cross-reaction between alphavirus species, the results were confirmed by  PRNT to avoid false positives (commission errors), as requirements to include these animals in that study, the horses must be not vaccinated against alphavirus or have been moved from premises before the samples were taken (omission errors).  In this study we are detecting IgG antibodies, positive animals were infected by VEEV, but we don´t know when exactly were infected, for example, we have positive and negative cases from the same area, with the same weather or variable conditions, but probably some of the negative animals are younger than the positive (older animals have more time to be exposed and infected by the virus), then if we use negative samples as absence data in other kind of model, we can commit omission errors. If a virus has a low prevalence, you need a big size sample to be sure that an area is absent of the disease, and this sampling is expensive, and we are prone to make omission errors [23], The rest of the location data came from a second serological study, based on the determination of IgM antibody by ELISA. When an animal presented any nervous symptoms a sera sample was taken and sent to the laboratory to determine the presence of IgM antibody against different arboviruses. IgM antibodies are detectable in acute infections for up to 6 weeks [24]. Though, IgM antibodies from an alphavirus response could last around 2 to 3 months [25]. At difference to IgG, IgM antibody seems to be more specific than IgG [11,26], and the positive sample was no confirmed by another test. However, in this study, the symptomatology could have been caused by other infectious agents or even a toxicological etiology, as happened in  2019 with 141 cases but only six were positive for VEEV, then the negative data did not correspond to a quit real absence data and could affect the predicting results. For these reasons the EMN based on presence, like Maxent, are a perfect tool to analyze our VEEV data, are sure and we can trust in the location of the positive cases but not is the case of the negative sample.

Comments:

# l. 31: which kind of arthropods?

R/ It was added the following text:   mainly mosquitoes [5,6] in line 32

# l. 37-38: this means increasing cases in low altitudes? Please rephrase and link to sentence before.

R/ The sentence was modified as follows in lines 49-51:

In that study, altitude <100m was the only variable considered a risk factor in the multivariate analysis, indicated that the lower altitude the higher IgG positives cases to VEEV [7], 

# l. 38: Passive surveillance of? Please clarify.

R/ the following text was modified lines 52-56:

An 11 years passive surveillance study finished in 2019, (in which there were no actively search of cases by the animal health authorities but the samples were sent to the laboratory voluntarily by the animal owners), showed the presence of positive cases every year with apparent peaks every 6 years, one observed in 2009 and the second in 2015 during the evaluated period. Altitude was associated with positive cases, consistent with the previous cross-sectional study [11],

# l. 39: how do you see the recurring pattern of six years if only analysing 11 years?

R/ You are right for that reason we wrote: “with apparent peaks every 6 years during the evaluated period”.  The first peak was observed in 2009 and the second in 2015 we are expecting an increasing of cases in 2021, according to our results these peaks could be related to la Niña phenomenon

We add:  one observed in 2009 and the second in 2015 in line 55

# l. 39-40: here temperature and altitude are relevant factors, previously only altitude? Please merge this statements.

R/ In both studies altitude was associate with VEEV positive cases, the second study has not been submitted to any journal yet, because we recently include the ENSO phenomena in the analyzes, considering that we observe a possible cycle of VEEV increase cases every 6 years, for this analysis temperature was not include due it was correlated to altitude, we determinate that there is 3.4 times more risk to have VEEV positive cases during la Niña than without la Niña phenomenon.

 We modify the text to merge those statements as follows in line 56:

Altitude was associated with positive cases, consistent with the previous cross-sectional study,

# l. 43: “could produce” à Please rephrase. e.g “can infect”

R/ can infect was used instead of could produce line 58:

# l. 49: I do not understand. You also have absence-data from the randomly selected horses?

R/ Maxent only use presence data, the goal is to estimate the potential distribution of a species, the absences caused by non- environmental factors must be avoided. [2] Here, absence data must come from environmental conditions that are known to be unsuitable for the species, being crucial to know with accuracy the location of the presence data

We only use presence data for the analysis, yes, we have negative data, please see lines 256-281, we trying to explain why we use only positive samples (presence) than negative samples (absence).

# l. 49: Please give a more comprehensive introduction on the methodological background of maxent.

R/ The following text was added, lines 110-124:

MaxEnt 3.3.3k   modeling program [16] was utilized to model the distribution of VEEV in Costa Rica based on previously obtained geographical locations. MaxEnt utilizes a maximum entropy algorithm to analyze the values of environmental layers. In other words, the program analyzes the variables and choose the ones of maximum entropy, i.e. the most unconstrained ones.  MaxEnt takes a list of species presence locations as input, as well as a set of environmental predictors (e.g. precipitation, temperature) across a user-defined landscape that is divided into grid cells. To obtain a solution MaxEnt maximizes the so-called gain function, a penalized maximum likelihood function, to find a model that can best differentiate presences from background locations. The gain is closely related to deviance, a measure of goodness of fit used in generalized additive and generalized linear models. It starts at 0 and increases towards an asymptote during the run. During this process, Maxent is generating a probability distribution over pixels in the grid, starting from the uniform distribution and repeatedly improving the fit to the data. Maxent isn’t directly calculating “probability of occurrence”. The probability it assigns to each pixel is typically very small, as the values must sum to 1 over all the pixels in the grid.  Then the program uses the presence of the variant of interest to produce a characteristic map that shows suitability values between 0 and 1 indicating high suitability versus low suitability of species presence, respectively, represented by a cloglog format. This range of values is depicted using a color gradient [17,18].

# l. 73: “and the virus genome was complete sequenced” à not important, delete

R/ it was deleted  lines 98-99

# l. 89-90: Random test percentage? Please rephrase this sentence.

R/ Lines 133-134 the sentence was modified as:

Of the 188 registers of presence the 25%, (47 registers), were used for testing porpoise (a threshold to make a binary prediction). 

# l. 89-90: Please also use the absence data from the study [5] to check for the suitability for absence prediction.

MaxEnt is a program for modelling species distributions from presence-only species records (Jane Elith, Steven J. Phillips, Trevor Hastie, Miroslav Dudík, Yung En Chee and Colin J. Yates. 2010. A statistical explanation of MaxEnt for ecologists. Diversity and Distributions, 1–15.

R/ Please see the previous answers where we explain why we only use in positive samples.

# l. 97-100: Are this the results of one run? If so, please repeat the Maxent runs with changing training and test points to calculate an average AUC.

R/ Yes, it was the result of one run, however, we analyzed the data with replicates and the model AUC not improved. 

# l. 105: “suitability of obtaining”? à unclear, please rephrase, do you mean suitability of transmission?

R/Lines 154-155, it was changed to:

Variables Mean Temperature of the coldest quarter and Precipitation of the driest quarter contribute 49% of the VEEV presence cases

# l. 136: to keep highest option? à unclear, please rephrase

R/ Lines 187-188, it was modified to:

to keep a high number of VEEV cases.

# l. 169: “suitableness”?

R/ Lines 232-233 it was modified by

To demonstrate the suitability of this map,  positive VEEV equine cases of the 2016 outbreak located in Talamanca Sixaola, (southeastern region of the country) were also represented in Figure 4

# l. 169: Please add all information relevant to understand all details the figures in the figure captions.

R/ Lines 223-231.  The caption was modified by

Figure 4. The main map represents the Maxent output, the black dots are the VEEV positive data used in the training analysis, while the violet dots are the VEEV positive data used in the test analysis. The colors in the map indicate the suitability of the VEEV presence, the warmest color (yellow to red) shows the places with more suitability of VEEV presence, (0.7 to 1) while the coldest colors (deep blue to light green shows the lower suitability of VEEV presence 0 to 0.69, see the scale color. The amplified region (Sixaola) south of the country shows the cases of VEEV in 2016, which were not part of the analyzed data in this study. The map in gray represents the location of Costa Rica in Central America.

Lines 242-245.  The caption was modified by

Figure 5. Maxent output map represented the suitability of VEEV outbreaks by each of the 82 Costa Rica´s counties. The label color in the map indicates the suitability of VEEV outbreaks or cases.

# l. 228: Cx. Tritaeniorhynchus

R/ The specie was corrected in the line  327

Cx. tritaeniorhynchus

# l. 239: Opened the door? Please rephrase.

R/Line 339  the word “opened the door”  was changed to “allows”

# l. 245-247: What is the current monitoring scheme?

R/ there is not a monitoring system at this moment, we believe that VEEV cases in humans are probably misdiagnosed by Dengue or other febrile diseases.

Reviewer 3 Report

General assessment

In “An Ecological niche model to estimate the potential presence of Venezuelan equine encephalitis virus in Costa Rica” León et al. use modelling software to investigate potential ecological niches of VEEV in Costa Rica with a view of predicting potential future outbreaks. They employ a wide range of bioclimatological variables combined with existing VEEV prevalence data to produce country-wide suitability data and maps for VEEV outbreaks. While the data has generated a clear environmental map of where evidence of VEEV should be investigated, I feel without several other ecological variables (vector prevalence the biggest example here) the data here is only narrow in scope.

Major Comments

One major critique of the manuscript is the lack of data other than bioclimatological variables. While it is touched upon in the discussion, given the importance of the vector species in VEEV transmission, I feel without incorporating data relating to competent arthropod vector prevalence across the regions examined the strength of the data presented is limited. Even if only limited data is available, I feel it should be incorporated into the models presented here. If the data is not available, or is not able to be collected, then the manuscript title should be modified to better describe the data as an “environmental niche model” rather than ecological.

Another critique of this paper is one of clarity. While the data analysis is sound and thorough, I feel the way it is presented in text could be better explained to help the reader. The most obvious example of this is the use of the variable name in the abstract, results, and figures 1 – 3 (axes and titles) rather than the variable description. If the actual names could be incorporated into the text it would stop the reader is constantly having to refer back to the table.

Minor Comments

Line 36: First use of m.a.s.l should be expanded.

Table 103: The vertical alignment of text in column 1/column 2 should be examined as it is currently slightly confusing (maybe make both top aligned?).

Figure 1A & 1B: I think the X-axis values are missing a decimal point (i.e., 10x to high?)

Figure 4 & 5: I’m unsure as to why these need to be 2 separate figures? Could they be combined as one figure? Also, the legends should be better explained as I’m not sure what the rectangles next to the county names are meant to represent?

Author Response

Major Comments

One major critique of the manuscript is the lack of data other than bioclimatological variables. While it is touched upon in the discussion, given the importance of the vector species in VEEV transmission, I feel without incorporating data relating to competent arthropod vector prevalence across the regions examined the strength of the data presented is limited. Even if only limited data is available, I feel it should be incorporated into the models presented here. If the data is not available, or is not able to be collected, then the manuscript title should be modified to better describe the data as an “environmental niche model” rather than ecological.

R/ The comments are quite accurate about to change the title,please see lines 1-4 it was modified as: An Environmental niche model to estimate the potential presence of Venezuelan equine encephalitis virus in Costa Rica.

Also the following text was added in the introduction, see lines 33-39: There are three fundamental factors to keep arboviruses circulating and producing outbreaks in a given geographic area: virus, vectors, and the host-reservoirs. These components must be present in the same place and at the same time for the effective transmission of pathogens to occur and many variables influence or affect each of these components. VEEV is closely related to its vectors, mainly mosquitoes, and, these, in turn, are intently related to weather conditions. Vectors are influenced by climatic variables such as temperature, humidity, and rain [6], as well as elevation [7]

Another critique of this paper is one of clarity. While the data analysis is sound and thorough, I feel the way it is presented in text could be better explained to help the reader. The most obvious example of this is the use of the variable name in the abstract, results, and figures 1 – 3 (axes and titles) rather than the variable description. If the actual names could be incorporated into the text it would stop the reader is constantly having to refer back to the table.

R/ The comment was also correct and the variable names were incorporate in the text

Minor Comments

Line 36: First use of m.a.s.l should be expanded.

R/ The abbreviation was expanded in lines 47-48

Table 103: The vertical alignment of text in column 1/column 2 should be examined as it is currently slightly confusing (maybe make both top aligned?).

R/ Line 148: The format of the table title was changed.

Figure 1A & 1B: I think the X-axis values are missing a decimal point (i.e., 10x to high?)

R/ Lines 163-168  The original  variables values were calculated in that way for example for Annual mean temperature the units are Degrees Celsius by 10,  this was explained in the text

 Figure 4 & 5: I’m unsure as to why these need to be 2 separate figures? Could they be combined as one figure? Also, the legends should be better explained as I’m not sure what the rectangles next to the county names are meant to represent?

R/ The rectangles next to the county name represent the potential risk of VEEV presence in that county for these reasons if we merge both figures, the figure will see overloaded,

The following test was added in lines 244-245:

The rectangles next to the county name represent the potential risk of VEEV presence in that county

Reviewer 4 Report

This study evaluated the use of the ecological niche modelling software, MaxEnt, in predicting areas of Costa Rica that are favorable for Venezuelan Equine Encephalitis Virus (VEEV). The authors use numerous climatological variables and altitude combined with training data using VEEV seropositives to establish significant variables and maps of Costa Rican VEEV hotspots. This manuscript contributes information to our knowledge on variables impacting VEEV occurrence and uses ecological niche modeling, which is an increasingly accepted tool for establishing the anticipated range of vectors and vector-borne diseases.

However, I am concerned that some important considerations are not being adequately addressed in the experimental design, which is reflected in the AUC of 0.8. The authors do state in the discussion that there are numerous reservoir species and that there may be other variables impacting them. Knowing this, why have the authors not sought vegetation and habitat datasets to contribute to the model in an attempt to improve the AUC? Spillovers of vector borne diseases such as this are often driven by changes in the reservoir host and vector communities. These variables should be included in the model to evaluate whether they improve the AUC. Alternatively, adequate justification for not including these data and additional acknowledgement of the limitation of the data in decision making should be added.

On a related note, many of the test points fall into green and blue regions on the map indicating poor suitability of the model and reflecting the relatively low AUC. This may be improved through adding missing data, but additional consideration should be given to this phenomenon and what it could mean for using this model for decision making moving forward. As stated in the introduction, there are numerous subtypes of VEEV-could the different subtypes need to be evaluated separately? Have any of the infected horses been moved between areas leading to positives in low likelihood areas? Are there too few training datapoints to successfully model this vector-borne disease in Costa Rica? Just a few considerations I had while trying to interpret the findings.

I additionally have a few itemized points that should be addressed below.

Introduction

Additional background information on this vector-borne disease transmission cycle should be included in this introduction. There is no information included on the mosquito species involved, the reservoir hosts, or even what role humans and horses actually play in the cycle (dead-end hosts). I feel that in order for readers to fully understand the variables used and the model output, they should be given a much broader explanation of what this system looks like.

Line 36: Please define m.a.s.l. in text

Line 50: change geographically to geographical

Lines 55-56: This purpose statement is a bit confusing and should be rewritten.

Methods:

Line 84-85: This should be rewritten for clarity. I recommend, “…produce a characteristic map that shows suitability values between 0 and 1 indicating high suitability versus low suitability of species presence, respectively. This range of values is depicted using a color gradient.”

Line 86: Please clarify that “cloglog” here refers to the suitability values above.

Line 92: It was originally unclear to me that the only difference between these models is the presence of the altitude variable. Please clarify that “only bioclimatic layers” includes all but 1 removed variable.

Results:

Figure 3 caption: Please clarify in this caption that the graphic on the left is the isolation model and the model on the right is the full model.

Figure 3 caption: Please clarify where the red bar depicting this limiting value for the variables is generated from.

Line 162: add “in” after presence

Figure 4: It is difficult to see the white training points on this map-especially in some of the yellow areas. I recommend changing the color of these points so they’re easier to see.

Discussion:

Lines 199-203: I’m unclear if some information is missing here. You say, “…while  and November had 29%...". Is there supposed to be another month listed there?

Line 207: In general, try to avoid starting a sentence with an abbreviated genus name. When you do abbreviate Aedes, it is typically abbreviated as Ae.

Line 228: Species names should be lowercase

Author Response

This study evaluated the use of the ecological niche modelling software, MaxEnt, in predicting areas of Costa Rica that are favorable for Venezuelan Equine Encephalitis Virus (VEEV). The authors use numerous climatological variables and altitude combined with training data using VEEV seropositives to establish significant variables and maps of Costa Rican VEEV hotspots. This manuscript contributes information to our knowledge on variables impacting VEEV occurrence and uses ecological niche modeling, which is an increasingly accepted tool for establishing the anticipated range of vectors and vector-borne diseases.

However, I am concerned that some important considerations are not being adequately addressed in the experimental design, which is reflected in the AUC of 0.8. The authors do state in the discussion that there are numerous reservoir species and that there may be other variables impacting them. Knowing this, why have the authors not sought vegetation and habitat datasets to contribute to the model in an attempt to improve the AUC? Spillovers of vector borne diseases such as this are often driven by changes in the reservoir host and vector communities. These variables should be included in the model to evaluate whether they improve the AUC. Alternatively, adequate justification for not including these data and additional acknowledgement of the limitation of the data in decision making should be added.

On a related note, many of the test points fall into green and blue regions on the map indicating poor suitability of the model and reflecting the relatively low AUC. This may be improved through adding missing data, but additional consideration should be given to this phenomenon and what it could mean for using this model for decision making moving forward. As stated in the introduction, there are numerous subtypes of VEEV-could the different subtypes need to be evaluated separately? Have any of the infected horses been moved between areas leading to positives in low likelihood areas? Are there too few training datapoints to successfully model this vector-borne disease in Costa Rica? Just a few considerations I had while trying to interpret the findings.

 R/ About the missing data or negative samples, Maxent only use presence data, and for that reason, we select this EMN, considering the kind of data that we have, in the case of the IgM study (animals with nervous symptoms), the symptomatology in some of these animals could have been caused by other infectious agent or even a toxicologic etiology, as happened in  2019, in that year the number of horse cases with nervous symptoms increased over the average, with receive 141 cases but only six were positive to VEEV, most of the cases were caused by other etiology.  Please, see information in lines 256-281 in the manuscript.

In relation, to your question … “there are numerous subtypes of VEEV-could the different subtypes need to be evaluated separately?” 

R/ Costa Rica is endemic to subtype IE, no other subtype was detected until now.

In line 45-46 we add the following text:

 also this study gave evidence that subtype IE is endemic in the country [2]

Have any of the infected horses been moved between areas leading to positives in low likelihood areas?

R/ The samples were obtained from two studies, in the first study, the distribution results came from a cross-sectional study done in 2013 based on the presence of IgG Aphavirus antibodies by ELISA however due to the IgG cross-reaction between alphavirus species, Venezuelan equine encephalitis, Eastern equine encephalitis, Western equine encephalitis, among others, the results were confirmed by plaque reduction neutralization test (PRNT), and the requirements to include these animals in that study were: not have been vaccinated against alphavirus or have been moved from a different site from where the sample was taken [1].  The rest of the location data came from a second study,  another serological study but a difference from the previous one is that the diagnostic was based on the determination of IgM antibody by ELISA,  IgM antibodies are detectable in acute infections for up to 6 weeks [2].  However, IgM antibodies from an alphavirus response could last around 2 to 3 months [3], the samples were sent to the laboratory at the moment the animals present nervous symptoms [4]. In both studies, VEEV was the virus more prevalent, and only data with coordinates were used for this analysis.

Please see the information added on lines 256-281.

Are there too few training data points to successfully model this vector-borne disease in Costa Rica?

R/ We don´t think so, we have 141 positive cases in the tranning model, there are articules published using MaxEnt with fewer data.

I additionally have a few itemized points that should be addressed below.

Introduction

Additional background information on this vector-borne disease transmission cycle should be included in this introduction. There is no information included on the mosquito species involved, the reservoir hosts, or even what role humans and horses actually play in the cycle (dead-end hosts). I feel that in order for readers to fully understand the variables used and the model output, they should be given a much broader explanation of what this system looks like.

R/ The following information was added in lines 33-43

There are three fundamental factors to keep arboviruses circulating and producing outbreaks in a given geographic area: virus, vectors, and the host-reservoirs. These components must be present in the same place and at the same time for the effective transmission of pathogens to occur and many variables influence or affect each of these components. VEEV is closely related to its vectors, mainly mosquitoes, and, these, in turn, are intently related to weather conditions. Vectors are influenced by climatic variables such as temperature, humidity, and rain [6], as well as elevation [7] or by less ecological aspects such as transport. from larvae or adults to new areas where they did not exist before through vehicles or objects [8]. The reservoirs and hosts maintain the viruses both in the enzootic cycles, as sporadically in the epizootic ones, aspects such as susceptibility, permissibility favor the replication of these viruses [9]

And also in lines 49-56

In that study, altitude <100m was the only variable considered a risk factor in the multivariate analysis, indicated that the lower altitude the higher IgG positives cases to VEEV [10]. An 11 years passive surveillance study finished in 2019, (in which there were no actively search of cases by the animal health authorities but the samples were sent to the laboratory voluntarily by the animal owners), showed the presence of positive cases every year with apparent peaks every 6 years, one observed in 2009 and the second in 2015 during the evaluated period. Altitude was associated with positive cases, consistent with the previous cross-sectional study [11],

Line 36: Please define m.a.s.l. in text

R/It was defined in line 48

Line 50: change geographically to geographical

R/ it was changed in line 71

Lines 55-56: This purpose statement is a bit confusing and should be rewritten.

 R/ It was rewritten in lines 74-80

Maxent is an ecological niche modeling program based on presence of species for this reason, it was selected to model the spatial data based on the serological presence of VEEV in horses to establish potential areas at risk for the appearance of this virus in Costa Rica using bioclimatic variables, and elevation data, The knowledge generated with this study could be important to help to prioritize resources, improve planning, prevention and response strategies to future surveillance and control programs for this virus.

Methods:

Line 84-85: This should be rewritten for clarity. I recommend, “…produce a characteristic map that shows suitability values between 0 and 1 indicating high suitability versus low suitability of species presence, respectively. This range of values is depicted using a color gradient.”

R/ We have accepted the recommendation, thank you, please see lines 117-123

Line 86: Please clarify that “cloglog” here refers to the suitability values above.

R/ it was already referred, please see the previous comment.  Lines 117-123

Line 92: It was originally unclear to me that the only difference between these models is the presence of the altitude variable. Please clarify that “only bioclimatic layers” includes all but 1 removed variable.

 R/ The text was modified, please see lines 135-136

1) all layers (bioclimatic and altitude variables) 2) only bioclimatic layers (altitude was removed) 

Results:

Figure 3 caption: Please clarify in this caption that the graphic on the left is the isolation model and the model on the right is the full model.

R/. Figure 3a, shows the test gain for VEEV (the effect of variables in the set of samples used to validate the model), while figure 3b shows the training gain for VEEV (the full model)

Figure 3 caption: Please clarify where the red bar depicting this limiting value for the variables is generated from.

R/ …the effect in the model when this variable is not considered. The red bar represents the performance of the model when all variables are included, if a blue light bar (the model is not using this variable) is longer than the red bar, means that the predictive performance of the model improves when the corresponding variable is not used. In our case, none variable

Line 162: add “in” after presence

R/ please see the change in line 222

Figure 4: It is difficult to see the white training points on this map-especially in some of the yellow areas. I recommend changing the color of these points so they’re easier to see.

 R/  The color of the training points were changed from white to black.

Discussion:

Lines 199-203: I’m unclear if some information is missing here. You say, “…while  and November had 29%...". Is there supposed to be another month listed there?

R/ The word ”and” was deleted in line 303 it can read it as:

while November had 29% of the positive cases, these months had the highest cases of VEEV during the studied period

Line 207: In general, try to avoid starting a sentence with an abbreviated genus name. When you do abbreviate Aedes, it is typically abbreviated as Ae.

R/  Thank you for the observation, ti was changed to Ae. Taeniorhynchus line  339

Line 228: Species names should be lowercase

R/ the change was done line 331

Reviewer 5 Report

Overall the study is well designed and data is properly analysed. The finding is useful on the the management of potential disease epidemic. However, there are a few places could be improved:

L21 'AUC' appears without definition;

L90 '25% of 188 data': where did the 188 come from, and what does 'data' mean here?

L97 'data' appears twice, and I can see 47 = 25% of 188

L99 'the climate variables were used': it seems good enough as compared 0.78 to 0.80; but how many other variables not in 'climate variables'?

What are the X-axis units in Figs 1 and 2?

Font too small in Fig 3

Is Fig 5 necessary? Will Fig 4 is sufficient if county boundaries are marked on Fig 4?

As mosquitoes are the main vector (as I understood), the virus epidemic is dependent on the availability of mosquitoes. Thus the factors influence mosquito population will definitely affect the virus. So the prediction factors could be further reduced as it is obviously the mean temperature of the coldest quarter will affect the annual mean temperature, and the precipitation  obviously affect mosquito reproduction. Simplify model imports would improve the usefulness of the potential forecasting model.

To improve the readability, remove some extra spaces between words.

Author Response

Comments and Suggestions for Authors

Overall the study is well designed and data is properly analysed. The finding is useful on the the management of potential disease epidemic. However, there are a few places could be improved:

L21 'AUC' appears without definition;

R/ area under the ROC curve was added  in line 21

L90 '25% of 188 data': where did the 188 come from, and what does 'data' mean here?

R/ in the lines 89-100 the following information was modified as follow:

We included two sets of VEEV positive data confirmed by two methodologies; the plaque reduction neutralization test (PRNT) and the IgM ELISA from two different serological studies. In the first study, 217 horses were randomly selected across the country in the lowlands and highlands, 81 horses were IgG positive by ELISA and confirmed by PRNT [10]. The second study comprised passive surveillance carried out between 2009 and 2019 involving samples from animals with neurological signs that were taken and sent to the laboratory at the moment the animals present symptomatology to detect the presence of IgM antibodies to VEEV and other arboviruses. One-hundred-and twenty-eight horses had IgM antibodies to VEEV but only 107 had geographic coordinate data [11]. One of these cases was also positive by RT-PCR, [14]. In both studies, VEEV was the virus more prevalent, and only data with coordinates were used for this analysis, Considering these conditions 188 registers of VEEV presence were utilized in this study

Also a modification was done in lines 132-133

Of the 188 registers of presence the 25%, (47 registers), were used for testing porpoise (a threshold to make a binary prediction). 

L97 'data' appears twice, and I can see 47 = 25% of 188

R/ In lines 141-142. The following text was deleted:

and 47 positive data were used to test and validate the model.

L99 'the climate variables were used': it seems good enough as compared 0.78 to 0.80; but how many other variables not in 'climate variables'?

R/ Sorry but the question was no clear for us, could you please explain … but how many other variables not in 'climate variables'?

What are the X-axis units in Figs 1 and 2?

R/  The following text was added in lines 162-167

 In figure 1a, Bio 11 Mean Temperature of the coldest quarter , the X-axis represents Degrees Celsius by 10, figure 1b Bio 17 Precipitation of the driest quarter the units are millimeters of precipitation, figure 1c, Bio 2 Annual mean temperature the units are Degrees Celsius by 10, figure 1d X-axis units are millimeters above sea level, and figure 1e units are millimeters of precipitation. These units are the same for figure 2  Font too small in Fig 3

R/ yes it is, but the figure has a good resolution  300dpi to enlarge it.

Is Fig 5 necessary? Will Fig 4 is sufficient if county boundaries are marked on Fig 4?

R/ Yes, it is necessary, because the legend in figure 5 indicates the risk that every county has, if we merge both pictures in one it will be quite overloaded due to labels. 

As mosquitoes are the main vector (as I understood), the virus epidemic is dependent on the availability of mosquitoes. Thus the factors influence mosquito population will definitely affect the virus. So the prediction factors could be further reduced as it is obviously the mean temperature of the coldest quarter will affect the annual mean temperature, and the precipitation  obviously affect mosquito reproduction. Simplify model imports would improve the usefulness of the potential forecasting model.

Simplify model imports would improve the usefulness of the potential forecasting model.

R/ We think that this type of modeling doesn't really work like that. There are no problems with the number of variables used, because the end goal is to create a predictive model. It does not work like other types of models where the principle of parsimony applies.

Thus the factors influence mosquito population will definitely affect the virus….

R/ You are right but besides mosquitoes,  reservoirs like rodents and some birds are also important to keep this virus circulating and the weather conditions could be different for these species. We try to eliminate some of these variables but the model did not improve the suitability to detect VEEV.

To improve the readability, remove some extra spaces between words.

R/ It was done, thank you for your advice

Round 2

Reviewer 1 Report

The authors have made significant revisions that enhance their manuscript. I have a few more minor comments for the authors to address.

"Environmental" in the title should be lower case.

I still suggest that the authors replace the "bio#" label in the graph titles in Figures 1 and 2 to the actual parameter. This information is now present in the figure legends, which helps, but could make the figures easier to read by writing "Response of VEEV to mean yemperature of the coldest quarter" instead of "bio11".  I also suggest making the A, B, C, D, E titles for each panel larger and moved them to the left side of each panel for clarity.

MaxEnt is written interchangeably throughout that manuscript also as "Maxent". Please choose one format and use throughout.

Author Response

Author's Reply to the Review Report (Reviewer 1)

Second round

Comments and Suggestions for Authors

The authors have made significant revisions that enhance their manuscript. I have a few more minor comments for the authors to address.

"Environmental" in the title should be lower case.

R/ the word was corrected line 2

I still suggest that the authors replace the "bio#" label in the graph titles in Figures 1 and 2 to the actual parameter. This information is now present in the figure legends, which helps, but could make the figures easier to read by writing "Response of VEEV to mean yemperature of the coldest quarter" instead of "bio11".  I also suggest making the A, B, C, D, E titles for each panel larger and moved them to the left side of each panel for clarity.

R/ The changes were done, please see the figures 1 and 2

MaxEnt is written interchangeably throughout that manuscript also as "Maxent". Please choose one format and use throughout.

R/ The word MaxEnt was homogenized through the text. 

Reviewer 3 Report

Authors have addressed all of my comments.

Author Response

Please, find the last version of the ms.

Reviewer 4 Report

This manuscript, documenting the use of MaxEnt software for determining VEEV presence in Costa Rica, has been improved significantly by the authors. While I still have some reservations about the lack of certain variables that have greater influence on the reservoir hosts of the virus, I do feel that the rest of my concerns were adequately addressed in the first round of revision. Specifically, additional wording was added to the introduction on the transmission cycle of this virus, clarification was added to the methods and results to aid in interpretation, and concerns about the number of training points and movement of horses were addressed. At this time, I feel that the manuscript is of sufficient scientific quality to warrant publication.

I do have some wording and editorial comments throughout, which are listed below.

Abstract (Line 15): Please put commas before and after, "depending on the subtype"

Abstract (Line 16): Replace "could be" with are

Abstract (Line 18): Replace "data of occurrence" with "occurrence data"

Introduction (Line 37): "Mainly mosquitoes" is redundant here with an earlier sentence, replace with "VEEV is closely related to its mosquito vectors and..."

Introduction (Line 40): Awkward wording, please consider changing to, "transportation of larvae or adults..."

Introduction (Line 41-43): This sentence is confusing and should be rewritten, perhaps as "The reservoirs and hosts maintain the viruses both in the enzootic and occasionally in the epizootic cycles. Host factors such as susceptibility and permissibility favor the replication of these viruses."

Introduction (Line 50): Edit to "indicating that higher IgG positive cases to VEEV were associated with lower altitudes"

Introduction (Lines 51-52): Change wording in parenthesis to "in which animal health authorities did not actively search for cases, but samples were sent to the laboratory voluntarily by animal owners"

Introduction (Line 68): Include a period after solution

Introduction (Line 75): Include a period after data

Introduction (Line 76): This sentence could be improved by adding "and" after resources

Methods (Line 126): remove "the" between presence and 25%

Methods (Line 126): change "porpoise" to "purposes"

Results (Line 182-183): Change to, "While the suitability of VEEV presence increases to 80% with increasing altitude, it drops as altitude reaches 1400 m.a.s.l. (Figure 1d)."

Results (Line 185-186): This opening sentence is redundant and can be largely removed. Can be rewritten as, "Precipitation of the warmest quarter is diluted in the presence of other variables..."

Results (Line 190): please add "the" between with and highest

Results (Line 192): put figure 3a in parenthesis

Results (Line 192): Change showing to shown

Results (Line 224): Remove the phrase "As it can see"

Results (Line 226): Change "Costa Rica's counties" to counties of Costa Rica

Discussion (Line 246): Alphavirus is misspelled

Discussion (Lines 244-249): This sentence is extremely long and should be broken up for clarity. I recommend ending the previous sentence after ELISA and starting a new sentence, "Due to the IgG cross-reaction between alphavirus species, the results were confirmed by PRNT to avoid false positives, or commission errors." 

Discussion (Line 251): add "they" before were infected

Discussion (Lines 249-254): This is another sentence that should be broken up into multiple sentences for clarity. I recommend adding a period after "infected" and start a new sentence with "For example, we have positive...". I would also advise adding a period after "by the virus)" and starting a new sentence with "If we use negative samples as absence data..."

Discussion (Line 254): change "other" to "another"

Discussion (Line 262): change "no" to "not"

Discussion (Line 265): remove "a quit"

Discussion (Line 267): The second half of this sentence is unclear, I would edit to, "our VEEV data where we are sure we can trust the positive cases but not necessarily the negative cases."

Discussion (Lines 272-273): I would include the line "Ebola virus presence AUC 0.9 or mosquitoes vector presence 0.94 and 0.9" in parentheses here.

Discussion (Line 290): Aedes is typically shortened to Ae. to avoid confusion with Anopheles

Discussion (Lines 291-293): Are these lines on precipitation of the driest quarter meant to be included here? They don't seem to fit with the rest of this paragraph.

Discussion (Lines 312-313): Recommend changing wording slightly to, "...for results reported for the vectors Cx. tritaeniorhynchus and Aedes albopictus."

Discussion (Line 316): Add a comma after "...probably the most significant"

Discussion (Line 321): Italicize the species name here.

Discussion (Line 327): Use a period after etiological agents and start a new sentence with, "In this study"

Discussion (Line 328): You don't need the commas before and after the statement in parentheses

Discussion (Line 328): Change "see" to "be seen"

Discussion (Line 330-332): Recommend adding this line to the end of the previous paragraph than having it standalone.

Discussion (Line 330): Add "that" after counties

Conclusions (Line 336): Remove the comma after "We hope this information"

Figures 5 and S1: The black dots used for figure 4 significantly improved the ability to interpret the map. I recommend switching from white dots to black dots for these other two figures as well.

Author Response

Author's Reply to the Review Report (Reviewer 4)

Second round

Review Report Form

Open Review

English language and style

( ) Extensive editing of English language and style required
(x) Moderate English changes required
( ) English language and style are fine/minor spell check required
( ) I don't feel qualified to judge about the English language and style

Yes

Can be improved

Must be improved

Not applicable

Does the introduction provide sufficient background and include all relevant references?

(x)

( )

( )

( )

Is the research design appropriate?

( )

(x)

( )

( )

Are the methods adequately described?

(x)

( )

( )

( )

Are the results clearly presented?

(x)

( )

( )

( )

Are the conclusions supported by the results?

(x)

( )

( )

( )

Comments and Suggestions for Authors

This manuscript, documenting the use of MaxEnt software for determining VEEV presence in Costa Rica, has been improved significantly by the authors. While I still have some reservations about the lack of certain variables that have greater influence on the reservoir hosts of the virus, I do feel that the rest of my concerns were adequately addressed in the first round of revision. Specifically, additional wording was added to the introduction on the transmission cycle of this virus, clarification was added to the methods and results to aid in interpretation, and concerns about the number of training points and movement of horses were addressed. At this time, I feel that the manuscript is of sufficient scientific quality to warrant publication.

I do have some wording and editorial comments throughout, which are listed below.

Abstract (Line 15): Please put commas before and after, "depending on the subtype"

R/ The change was done ´please see line 15

Abstract (Line 16): Replace "could be" with are

R/ The word are was used intead could be, line 16

Abstract (Line 18): Replace "data of occurrence" with "occurrence data"

R/Change was done line 18

Introduction (Line 37): "Mainly mosquitoes" is redundant here with an earlier sentence, replace with "VEEV is closely related to its mosquito vectors and..."

R/ Change was done in line 37

Introduction (Line 40): Awkward wording, please consider changing to, "transportation of larvae or adults..."

R/ thank you, the change was done, line 40

Introduction (Line 41-43): This sentence is confusing and should be rewritten, perhaps as "The reservoirs and hosts maintain the viruses both in the enzootic and occasionally in the epizootic cycles. Host factors such as susceptibility and permissibility favor the replication of these viruses."

R/ Thank you, the text was modified as was suggested, please see lines 41-43

Introduction (Line 50): Edit to "indicating that higher IgG positive cases to VEEV were associated with lower altitudes"

Introduction (Lines 51-52): Change wording in parenthesis to "in which animal health authorities did not actively search for cases, but samples were sent to the laboratory voluntarily by animal owners"

R/ The text was modified, please see lines 52-53

Introduction (Line 68): Include a period after solution

R/ Done, line 69

Introduction (Line 75): Include a period after data

R/Done, line 77

Introduction (Line 76): This sentence could be improved by adding "and" after resources

R/Done, line 78.

Methods (Line 126): remove "the" between presence and 25%

R/Done, line 129

Methods (Line 126): change "porpoise" to "purposes"

R/  Thank you, I did not see it, the change was done.

Results (Line 182-183): Change to, "While the suitability of VEEV presence increases to 80% with increasing altitude, it drops as altitude reaches 1400 m.a.s.l. (Figure 1d)."

R/ Change was done, lines 187-188

Results (Line 185-186): This opening sentence is redundant and can be largely removed. Can be rewritten as, "Precipitation of the warmest quarter is diluted in the presence of other variables..."

R/ The change was done, lines 192-193

Results (Line 190): please add "the" between with and highest

R/ The word was added, line 197

Results (Line 192): put figure 3a in parenthesis

R/  The change was done, line 199

Results (Line 192): Change showing to shown

R/The change was done, line 199

Results (Line 224): Remove the phrase "As it can see"

R/ The change was done, line 231

Results (Line 226): Change "Costa Rica's counties" to counties of Costa Rica

R/ The change was done, line 233-234

Discussion (Line 246): Alphavirus is misspelled

R/ Thank you, the word was corrected, line

Discussion (Lines 244-249): This sentence is extremely long and should be broken up for clarity. I recommend ending the previous sentence after ELISA and starting a new sentence, "Due to the IgG cross-reaction between alphavirus species, the results were confirmed by PRNT to avoid false positives, or commission errors." 

R/ The suggestion was accepted, please see line 253

Discussion (Line 251): add "they" before were infected

R/ The change was done, line 258

Discussion (Lines 249-254): This is another sentence that should be broken up into multiple sentences for clarity. I recommend adding a period after "infected" and start a new sentence with "For example, we have positive...". I would also advise adding a period after "by the virus)" and starting a new sentence with "If we use negative samples as absence data..."

R/ Thank you for your valuable suggestion, the change was done, lines 258 and 261

Discussion (Line 254): change "other" to "another"

R/The change was done, line 261

Discussion (Line 262): change "no" to "not"

R/Change was done, line 270

Discussion (Line 265): remove "a quit"

The words were removed, line 272

Discussion (Line 267): The second half of this sentence is unclear, I would edit to, "our VEEV data where we are sure we can trust the positive cases but not necessarily the negative cases."

R/ The change was done, lines 274-275

Discussion (Lines 272-273): I would include the line "Ebola virus presence AUC 0.9 or mosquitoes vector presence 0.94 and 0.9" in parentheses here.

R/ Parentheses were added at lines 280-281.

Discussion (Line 290): Aedes is typically shortened to Ae. to avoid confusion with Anopheles

R/ The change was done,  see please, line 298.

Discussion (Lines 291-293): Are these lines on precipitation of the driest quarter meant to be included here? They don't seem to fit with the rest of this paragraph.

R/Thank you to noted this. The text was deleted, lines 300-301

Discussion (Lines 312-313): Recommend changing wording slightly to, "...for results reported for the vectors Cx. tritaeniorhynchus and Aedes albopictus."

R/ The text was modified, lines 320-321

Discussion (Line 316): Add a comma after "...probably the most significant"

R/ A comma was added, line 324

Discussion (Line 321): Italicize the species name here.

R/ The modification was done, please see line 329

Discussion (Line 327): Use a period after etiological agents and start a new sentence with, "In this study"

R/The change was done line 334

Discussion (Line 328): You don't need the commas before and after the statement in parentheses

R/Commas were deleted in line 336

Discussion (Line 328): Change "see" to "be seen"

R/ The change was done, line 337

Discussion (Line 330-332): Recommend adding this line to the end of the previous paragraph than having it standalone.

R/The text was modified as follow in line 337

… the enlarged image (Figure 5). The map in Figure 5, could be used in…

Discussion (Line 330): Add "that" after counties

R/ This text was modified, please see line 337

Conclusions (Line 336): Remove the comma after "We hope this information"

R/The comma was deleted, line 343

Figures 5 and S1: The black dots used for figure 4 significantly improved the ability to interpret the map. I recommend switching from white dots to black dots for these other two figures as well.

R/ The maps of these figures were changed,
